# Association between Fibrinogen-to-Albumin Ratio and Prognosis in Patients Admitted to an Intensive Care Unit

**DOI:** 10.3390/jcm12041407

**Published:** 2023-02-10

**Authors:** Keun-Soo Kim, Ah-Ran Oh, Jungchan Park, Jeong-Am Ryu

**Affiliations:** 1Department of Critical Care Medicine, Samsung Medical Center, Sungkyunkwan University School of Medicine, Seoul 06351, Republic of Korea; 2Department of Anesthesiology and Pain Medicine, Samsung Medical Center, Sungkyunkwan University School of Medicine, Seoul 06351, Republic of Korea; 3Department of Neurosurgery, Samsung Medical Center, Sungkyunkwan University School of Medicine, Seoul 06351, Republic of Korea

**Keywords:** fibrinogen, albumin, SOFA score, prognosis, intensive care unit

## Abstract

The objective of this study was to investigate the usefulness of fibrinogen-to-albumin ratio (FAR) as a prognostic marker in patients admitted to an intensive care unit (ICU) compared with Sequential Organ Failure Assessment (SOFA) score, a widely used prognostic scoring system. An inverse probability weighting (IPW) was used to control for selection bias and confounding factors. After IPW adjustment, the high FAR group showed significantly higher risk of 1-year compared with low FAR group (36.4% vs. 12.4%, adjust hazard ratio = 1.72; 95% confidence interval (CI): 1.59–1.86; *p* < 0.001). In the receiver-operating characteristic curve analysis associated with the prediction of 1-year mortality, there was no significant difference between the area under the curve of FAR on ICU admission (C-statistic: 0.684, 95% CI: 0.673–0.694) and that of SOFA score on ICU admission (C-statistic: 0.679, 95% CI: 0.669–0.688) (*p* = 0.532). In this study, FAR and SOFA score at ICU admission were associated with 1-year mortality in patients admitted to an ICU. Especially, FAR was easier to obtain in critically ill patients than SOFA score. Therefore, FAR is feasible and might help predict long-term mortality in these patients.

## 1. Introduction

Predicting the prognosis of critically ill patients can help maintain adequate management and prevent futile treatment in the intensive care unit (ICU) [1]. However, it is not easy to predict the prognosis of patients admitted to an ICU at the early stage [2]. Generally, prognostic scoring systems and biomarkers have been commonly used for predicting clinical outcomes in critically ill patients. They can help us assess the severity of disease, cost-benefit analysis, and clinical decision-making [3,4]. However, most scoring systems are difficult to obtain and complex in critically ill patients. Sequential Organ Failure Assessment (SOFA) score is a well-known scoring system. SOFA score can help us assess organ dysfunction or failure. They are useful for predicting clinical outcomes in patients admitted to an ICU [5].

Serum biomarkers are easy to obtain and known to be helpful for making early decisions in ICU treatment [6,7]. Among numerous biomarkers, fibrinogen and serum albumin can be used as predictors of morbidity and mortality in critically ill patients [8,9,10,11]. Fibrinogen, a human plasma glycoprotein synthesized in the liver, plays a role in the coagulation pathway as a clotting factor. Fibrinogen is also involved in tissue injury and inflammation [8,11,12,13]. In addition, fibrinogen is associated with prognosis of several diseases such as sepsis, coronary artery disease, diabetes, and chronic kidney disease [14,15]. In addition, serum albumin reflects the nutritional status, degree of inflammation, and therapeutic effect [16]. Fibrinogen-to-albumin ratio (FAR) has been shown to be effective in reflecting coagulation and nutritional status as well as inflammatory status [16]. Recently, some studies have shown its usefulness and significance as a prognostic factor in several diseases including cancer, coronary artery disease, stroke-associated pneumonia, and sepsis [8,11,12,13]. However, studies about the relationship between initial FAR and clinical outcome of patients who are admitted to ICU are limited. Therefore, the objective of this study was to investigate the usefulness of FAR as a prognostic marker in patients admitted to an ICU for various causes compared with Sequential Organ Failure Assessment (SOFA) score, a widely used prognostic scoring system.

## 2. Materials and Methods

The Institutional Review Board of Samsung Medical Center approved this study (SMC 2022-07-050). Written informed consent from an individual patient was not required because this study used a de-identified registry. This study was conducted according to the Declaration of Helsinki. Results are reported following the Strengthening the Reporting of Observational Studies in Epidemiology guidelines.

### 2.1. Data Curation & Study Population

The registry had medical data of 65,654 consecutive adult patients who were treated at the ICU with SOFA score evaluated upon admission in Samsung Medical Center, Seoul, Korea between June 2013 and May 2022. In this study, the SOFA score was reported in the medical record for all included patients. This large, single-center cohort was generated in a de-identified form using data extracted by the institutional electronic archive system. The “Clinical Data Warehouse Darwin-C” is an electronic system built for investigators to search and retrieve data from institutional electronic medical records for over 4 million patients with more than 900 million laboratory findings and 200 million prescriptions. For mortality data outside our institution, this system uses a unique personal identification number updated from the National Population Registry of the Korea National Statistical Office. Using an extracted raw medical record, independent investigators who were blinded to mortality organized relevant variables of demographic data and underlying diseases. Results of blood laboratory tests and SOFA score were automatically extracted.

For this study, we selected patients with available fibrinogen and albumin levels at admission to ICU. The decision to measure fibrinogen levels was based on the individual patient’s clinical situation. Fibrinogen was routinely checked on ICU admission in patients with sepsis, suspicious disseminated intravascular coagulation, solid and hematologic malignancies, post-operative care, suspicious bleeding tendency, etc. We initially stratified study patients into three groups according to tertile values of FAR upon ICU admission. After estimating an optimal threshold associated with 1-year mortality, we divided patients accordingly.

### 2.2. Definitions & Study Endpoints

FAR was defined as the ratio of fibrinogen concentration (mg/dL) to albumin concentration (g/L). SOFA score was estimated by assessing each component of respiratory, coagulation, liver, cardiovascular, central nervous system, and renal parameters as described in a previous study [17]. In the case of multiple measurements of SOFA score, the worst value during the initial 24 h after ICU admission was used for analysis. The primary endpoint was mortality during 1-year follow-up. The secondary endpoints were 30-day mortality, in-hospital mortality, and ICU mortality.

### 2.3. Statistical Analysis

Continuous variables are presented as means ± standard deviations and categorical variables are represented as numbers with subsequent percentages. Data were compared using Student’s *t*-test and one-way analysis of variance for continuous variables and Chi-square test or Fisher’s exact test for categorical variables. To estimate an optimal threshold of FAR associated with 1-year mortality, we generated receiver-operating characteristic (ROC) plots. We also determined the specificity and sensitivity of the threshold. We obtained optimal cut-off for FAR to predict mortality during 1-year follow-up based on the ROC curve and Youden index [18,19]. Mortalities were compared according to the estimated threshold using Cox regression analysis. Results are presented as hazard ratio (HR) with a 95% confidence interval (CI). A rigorous statistical adjustment was conducted to reduce bias and achieve a balance between groups. We chose an inverse probability weighting (IPW) using propensity score and adjusted for all relevant variables [20]. We compared the balance of baseline covariates between nutrition groups by calculating the absolute standardized difference (ASD) [21]. If IPW methods were effective for balancing exposure groups, the ASD should be close to zero [22]. The ASD under 10% was deemed as an adequate balance between groups. We also generated a Kaplan–Meier curve and compared the risk of mortality with log-rank test. The power of analysis was computed based on the sample size. The power of our analysis was 0.99 when HR was over 1.2. It was 0.78 when HR was 1.1 [23]. We assessed predictive performances of FAR and SOFA score in ICU admission using areas under the curve (AUCs) of the ROC curves for sensitivity vs. 1-specificity. We compared AUCs using the nonparametric approach published by DeLong et al. [24] for two correlated AUCs. All statistical analyses in this study were performed using R 4.2.0 (Vienna, Austria; http://www.R-project.org/ (accessed on 7 December 2021)).

## 3. Results

From 65,654 patients in the entire registry, we excluded 46,608 patients without available fibrinogen level. We then excluded 484 patients without available albumin level. A total of 18,562 (28.3%) patients were included for analysis. Tertile values of FAR were 6.41 and 10.59, respectively. Baseline characteristics of study patients according to the tertile values of FAR are summarized in Table 1. There were statistical differences in most variables between 3three groups, except severe trauma. Mortalities were higher in patients with higher FAR. Figure 1 shows changes of risk of 1-year mortality according to FAR upon ICU admission.

The optimal cutoff threshold of FAR associated with 1-year mortality was estimated to be 10.79 with the area under the ROC curve of 0.68. The sensitivity and specificity of the estimated threshold were 58.6% and 74.1%, respectively. According to this threshold, 12,533 (67.5%) and 6029 (32.5%) patients were stratified into a low FAR group and a high FAR group, respectively. The mean value of FAR was 6.57 in the low FAR group and 17.74 in the high FAR group. The high FAR group tended to be older, with higher incidences of comorbidities (Table 2). The risk of 1-year mortality was increased for the high FAR group compared with the low FAR group (36.4% vs. 12.4%; HR: 3.47; 95% CI: 3.25–3.70; *p* < 0.001 for 1-year mortality). Thirty-day mortality, in-hospital mortality, and ICU mortality were also higher in the high FAR group (19.1% vs. 7.6%; HR, 2.68; 95% CI, 2.46–2.92; *p* < 0.001 for 30-day mortality, 21.8% vs. 8.8%; HR, 2.74; 95% CI, 2.52–2.96; *p* < 0.001 for in-hospital mortality, and 11.2% vs. 4.9%; HR, 2.42; 95% CI, 2.17–2.70; *p* < 0.001 for ICU mortality). After adjustment with IPW technique, the association between increased risk of mortality and high FAR remained significant except for ICU mortality (HR, 1.72; 95% CI, 1.59–1.86; *p* < 0.001 for 1-year mortality, HR, 1.32; 95% CI, 1.19–1.46; *p* < 0.001 for 30-day mortality, HR, 1.33; 95% CI, 1.21–1.47; *p* < 0.001 for in-hospital mortality) (Table 3). The Kaplan–Meier curve for 1-year mortality according to FAR on ICU admission is shown in Figure 2. We also found that the association between FAR and mortality was significant regardless of the type of ICU (Figure 3).

In the ROC curve analysis associated with the prediction of 1-year mortality, there was no significant difference between the AUC of FAR on ICU admission (C-statistic: 0.684, 95% CI: 0.673–0.694) and that of SOFA score on ICU admission (C-statistic: 0.679, 95% CI: 0.669–0.688) (*p* = 0.532). The performance of SOFA score for predicting 1-year mortality was better than that of FAR in medical, oncologic, and neurosurgical ICUs, whereas FAR performed better than SOFA score in surgical ICU (Figure 3).

## 4. Discussion

In this study, we investigated the usefulness of FAR as a prognostic marker in patients admitted to an ICU and compared the predictive value with SOFA score. One-year mortality was higher in patients with higher FAR than in those with low FAR. After an adjustment with IPW technique, the association between increased risk of mortality and high FAR remained significant. There was no difference between prognostic value of FAR and that of SOFA score on ICU admission. However, FAR is easier and simpler to obtain than SOFA score in patients admitted to an ICU. Therefore, our results suggest that FAR on ICU admission is a feasible and useful marker to predict 1-year mortality in critically ill patients.

Predicting prognosis plays an important role in the management and decision-making of critically ill patients [25]. Appropriate treatment strategies might be optimized based on predicted prognosis of an individual patient [25]. Prognostic factors may prevent futile treatment and allow us to be more selective and focused on critically ill patients [26]. Additionally, patients and their families can be informed about the risk of recurrence or death. However, even a prognostic factor with an excellent predictive performance may not be feasible in a clinical practice if it is too difficult to obtain it. The SOFA score is a well-known prognostic tool that is widely used. In ICU stetting, it can predict the prognosis through multi-organ dysfunction. However, it is somewhat complex in that it is necessary to evaluate multiple organs. In contrast, FAR has the advantage in clinical practice because it is easily and readily measured through blood tests, so it may be useful for identifying patients who are at high risk of death and require aggressive treatment in the early stages of ICU admission.

Systemic inflammation and coagulation processes are closely related. Coagulation factors play an important role in the immune system [27]. Beside clotting and thrombosis, fibrinogen is closely associated with inflammatory reaction [14]. Especially, fibrinogen can modulate the immune system and act as a chemotactic factor for monocytes and neutrophils [28]. Therefore, fibrinogen can be an important factor in the cascade of inflammatory reactions. In addition, elevated fibrinogen levels would be more likely to lead to an increased risk of long-term mortality associated with underlying diseases than systemic organ dysfunctions [29]. A recent study has revealed that fibrinogen elevation is associated with excessive inflammation and disease severity in COVID-19 patients [30]. Serum albumin is also an indicator that reflects nutritional and inflammations [16]. While SOFA score is focused on organ dysfunction of critically ill patients, FAR could reflect inflammation, immune reaction, coagulation, and nutritional state related to underlying diseases which might be associated with long-term mortality. In this study, predictive performances of FAR and SOFA score on ICU admission for long-term outcomes of critically ill patients were found to be similar.

We also conducted separate analyses according to types of ICU. SOFA score was shown to outperform FAR in medical and oncologic ICU, which might be related to the fact that many patients were already diagnosed with sepsis during treatment of underlying diseases before ICU admission. Organ failure is more likely to be accompanied in critically ill patients with sepsis. Therefore, SOFA score on ICU admission might be useful for predicting prognosis in these patients [5]. In neuro-critically ill patients, FAR also showed a limited predictive value as an early prognostic factor. However, it is difficult to conclude this result owing to the small number of neurosurgical patients included in the study. FAR showed a significant association with 1-year mortality of surgical patients. This might be due to perioperative bleeding and transfusion that might be directly related to FAR [31].

In this study, only patients with fibrinogen available at ICU admission were included. This could raise concerns about selection bias and external validity of our findings. Additionally, they may have had more severe and complex disease, potentially leading to an overestimation of mortality. Nevertheless, our results showed consistent trend with previous studies, which demonstrated an association between high FAR and mortality risk in ICU patients [32,33]. Future studies with larger sample size, and with measurements of fibrinogen in all patients could provide further support for our findings.

Although we have suggested that FAR may be useful for ICU patients, indications for its use have not yet been established. It may not be necessary to measure fibrinogen and albumin in all patients upon ICU admission, as this would likely be resource-intensive and may not provide significant additional prognostic information for all patients. Instead, it may be more appropriate to use a risk stratification approach to identify patients at high risk of poor outcomes who would most benefit from measuring these biomarkers. Additionally, measuring fibrinogen and albumin in patients who have already been identified as being at high risk of poor outcomes based on other clinical or laboratory parameters may also be useful in predicting prognosis.

This study has several limitations. First, this was a retrospective review of medical records and data extracted from the Clinical Data Warehouse. The nonrandomized nature of registry data might have resulted in a selection bias. Second, laboratory tests including fibrinogen and albumin levels were not protocol-based for patients admitted ICU. Third, although fibrinogen and albumin levels were based on initially identified values to reduce the changes due to blood transfusions and albumin infusions after ICU admission, the values might have been affected in some patients. Fourth, in neuroscience ICU, FAR values could only be obtained for a few patients upon ICU admission. Finally, follow-up FAR data were obtained for a few patients. Therefore, it was unclear whether changes of FAR were associated with long-term mortality. Despite these limitations, this is the first study to evaluate the usefulness of FAR in the entire ICU. Additionally, our study differed from other studies in that FAR was analyzed relative to SOFA scores and analyzed separately for each ICU. In this context, the present study provides valuable insights, but large-scale prospective studies are needed to further confirm the usefulness of FAR in predicting clinical outcomes of critically ill patients with evidence-based conclusions.

## 5. Conclusions

In this study, FAR and SOFA score at ICU admission were associated with 1-year mortality in patients admitted ICU. However, obtaining FAR is easier than obtaining SOFA score in critically ill patients. Therefore, FAR is feasible and might help predict long-term mortality in these patients.

## Figures and Tables

**Figure 1 jcm-12-01407-f001:**
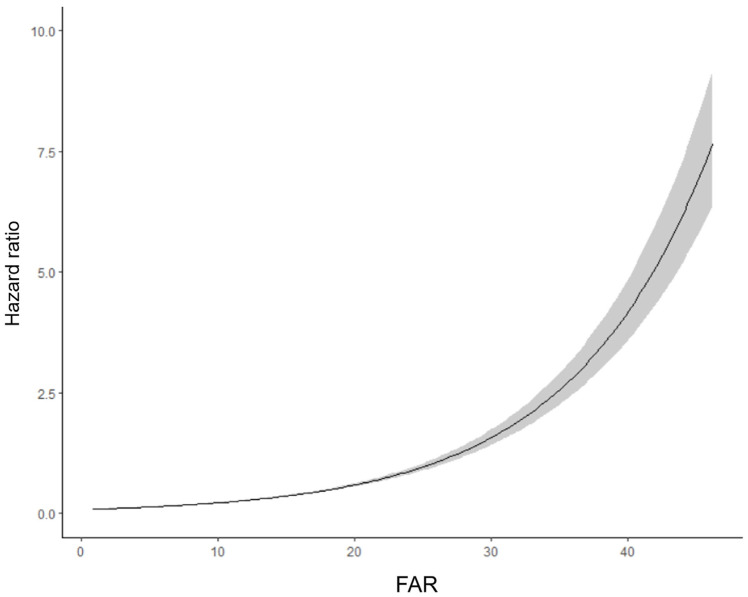
Estimated risk of one-year mortality according to fibrinogen/albumin ratio (FAR) upon intensive care unit admission.

**Figure 2 jcm-12-01407-f002:**
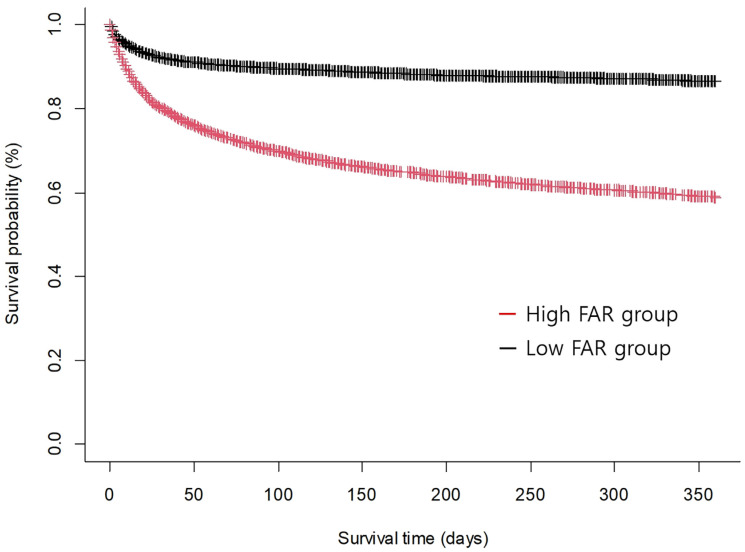
Kaplan–Meier curves for 1-year mortality according to fibrinogen/albumin ratio (FAR) upon intensive care unit admission (hazard ratio, 3.47; 95% confidence interval, 3.25–3.70; *p* < 0.001).

**Figure 3 jcm-12-01407-f003:**
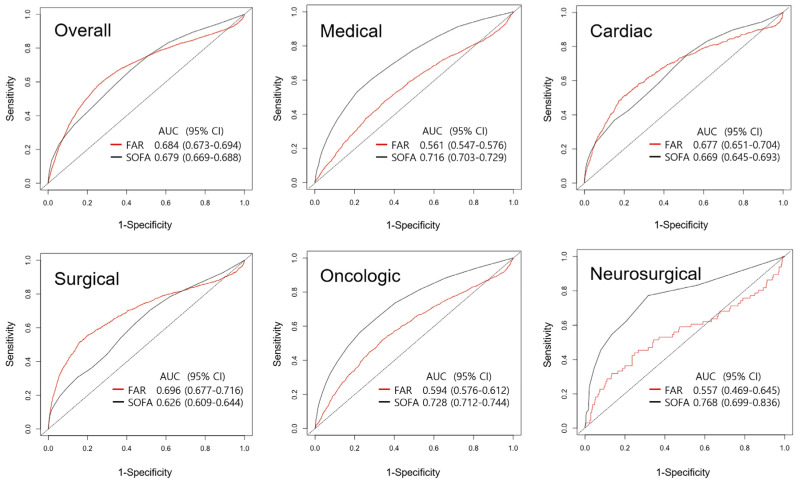
Receiver operating characteristic curves for prediction of 1-year mortality using fibrinogen/albumin ratio (FAR) and Sequential Organ Failure Assessment (SOFA) score. AUC, areas under the curve; CI, confidence interval.

**Table 1 jcm-12-01407-t001:** Baseline characteristics and mortalities according to tertile values of fibrinogen/albumin ratio.

	1st Tertile < 6.41 (*n* = 6189)	2nd Tertile < 10.59 (*n* = 6182)	3rd Tertile (*n* = 6191)	*p*-Value
Fibrinogen/albumin ratio	4.9 ± 1.1	8.1 ± 1.2	17.6 ± 6.3	<0.001
Fibrinogen, mg/dL	163.6 ± 48.7	259.5 ± 62.8	494.0 ± 161.6	<0.001
Albumin, g/L	33.1 ± 6.5	32.0 ± 6.6	28.8 ± 5.9	<0.001
SOFA score at ICU admission	5.4 ± 3.9	4.4 ± 3.8	4.7 ± 3.9	<0.001
Apache score at ICU admission	17.1 ± 5.7	15.5 ± 6.4	15.1 ± 6.8	<0.001
Age	57.6 ± 14.3	62.5 ± 13.5	63.2 ± 14.6	<0.001
Male	3788 (61.2)	4033 (65.2)	3956 (63.9)	<0.001
Comorbidities				
Hypertension	3804 (61.2)	4048 (65.3)	3925 (63.8)	<0.001
Malignancy	532 (8.6)	794 (12.8)	1233 (19.9)	<0.001
Diabetes mellitus	1368 (22.1)	1920 (31.1)	1972 (31.9)	<0.001
Chronic kidney disease	133 (2.1)	314 (5.1)	406 (6.6)	<0.001
Chronic liver disease	975 (15.8)	391 (6.3)	295 (4.8)	<0.001
Chronic obstructive pulmonary disease	151 (2.4)	191 (3.1)	235 (3.8)	<0.001
Stroke	289 (4.7)	410 (6.6)	356 (5.8)	<0.001
Heart failure	131 (2.1)	212 (3.4)	161 (2.6)	<0.001
Tuberculosis	204 (3.3)	271 (4.4)	422 (6.8)	<0.001
Coronary artery disease	351 (5.7)	663 (10.7)	547 (8.8)	<0.001
Habitual risk factors				
Current smoker	738 (11.9)	802 (13.0)	707 (11.4)	0.026
Alcohol intake	1517 (24.5)	1437 (23.2)	1132 (18.3)	<0.001
Cause of ICU admission				
Severe trauma	33 (0.5)	20 (0.3)	23 (0.4)	0.160
Perioperative management	4641 (75.0)	3948 (63.9)	1758 (28.4)	<0.001
Post-cardiac arrest syndrome	74 (1.2)	87 (1.4)	122 (2.0)	0.001
Neurological disorder	101 (1.6)	144 (2.3)	121 (2.0)	0.020
Respiratory distress	212 (3.4)	403 (6.5)	1821 (29.4)	<0.001
Cardiovascular disease	573 (9.3)	1048 (17.0)	1407 (22.7)	<0.001
Abdominal disorder	264 (4.3)	159 (2.6)	238 (3.8)	<0.001
Others	291 (4.7)	373 (6.0)	701 (11.3)	<0.001
ICU management				
Mechanical ventilation	2417 (39.1)	2252 (36.4)	2811 (45.4)	<0.001
Continuous renal replacement therapy	386 (6.2)	346 (5.6)	574 (9.3)	<0.001
Extracorporeal membrane oxygenation	397 (6.4)	387 (6.3)	496 (8.0)	<0.001
Use of vasopressor	2786 (45.0)	2300 (37.2)	2138 (34.5)	<0.001
Clinical outcomes				
One-year mortality	713 (11.5)	820 (13.3)	2213 (35.7)	<0.001
30-day mortality	499 (8.1)	447 (7.2)	1161 (18.8)	<0.001
Length of hospital stay, hour	560 (9.0)	527 (8.5)	1324 (21.4)	<0.001
During ICU stay, hour	344 (5.6)	268 (4.3)	679 (11.0)	<0.001

Data are presented as *n* (%) or mean (± standard deviation). SOFA = Sequential Organ Failure Assessment; ICU = intensive care unit.

**Table 2 jcm-12-01407-t002:** Baseline characteristics according to the estimated threshold of fibrinogen/albumin ratio of 10.79.

	Low Group (*n* = 12,533)	High Group (*n* = 6029)	Before IPW		After IPW
			*p*-Value	ASD	ASD
Fibrinogen/albumin ratio	6.6 ± 2.0	17.7 ± 6.2			
Fibrinogen, mg/dL	213.0 ± 74.9	498.4 ± 161.1			
Albumin, g/L	323.6 ± 6.6	28.8 ± 5.8			
SOFA score at ICU admission	4.9 ± 3.9	4.8 ± 3.9	0.010	4.2	2.6
Apache score at ICU admission	16.3 ± 6.1	16.0 ± 6.9	<0.001	18.0	7.9
Age	60.1 ± 14.1	63.2 ± 14.6	<0.001	21.3	0.2
Male	7929 (63.3)	3848 (63.8)	0.470	1.2	0.6
Comorbidities					
Hypertension	5333 (42.6)	2824 (46.8)	<0.001	8.6	1.5
Malignancy	1352 (10.8)	1207 (20.0)	<0.001	25.8	0.5
Diabetes mellitus	3341 (26.7)	1919 (31.8)	<0.001	11.4	1.6
Chronic kidney disease	456 (3.6)	397 (6.6)	<0.001	13.4	0.1
Chronic liver disease	1373 (11.0)	288 (4.8)	<0.001	23.1	1.3
Chronic obstructive pulmonary disease	347 (2.8)	230 (3.8)	<0.001	5.9	0.3
Stroke	709 (5.7)	346 (5.7)	0.850	0.4	<0.1
Heart failure	349 (2.8)	155 (2.6)	0.430	1.3	0.2
Tuberculosis	483 (3.9)	414 (6.9)	<0.001	13.4	0.4
Coronary artery disease	1038 (8.3)	523 (8.7)	0.380	1.4	1.8
Habitual risk factors					
Current smoker	1557 (12.4)	690 (11.4)	0.060	3.0	2.8
Alcohol intake	2988 (23.8)	1098 (18.2)	<0.001	13.8	3.2
Cause of ICU admission					
Severe trauma	53 (0.4)	23 (0.4)	0.770	0.7	0.1
Perioperative management	8686 (69.3)	1661 (27.6)	<0.001	0.9	2.7
Post-cardiac arrest syndrome	162 (1.3)	121 (2.0)	<0.001	5.6	0.6
Neurological disorder	249 (2.0)	117 (1.9)	0.880	0.3	1.2
Respiratory distress	628 (5.0)	1808 (30.0)	<0.001	69.6	0.3
Cardiovascular disease	1655 (13.2)	1373 (22.8)	<0.001	25.1	2.3
Abdominal disorder	426 (3.4)	235 (3.9)	0.090	2.7	0.2
Others	674 (5.4)	691 (11.5)	<0.001	22.0	0.6
ICU management					
Mechanical ventilation	4720 (37.7)	2760 (45.8)	<0.001	16.5	0.4
Continuous renal replacement therapy	742 (5.9)	564 (9.4)	<0.001	13.0	0.5
Extracorporeal membrane oxygenation	793 (6.3)	487 (8.1)	<0.001	6.8	1.9
Use of vasopressor	5138 (41.0)	2086 (34.6)	<0.001	13.2	2.8

Data are presented as *n* (%) or mean (±standard deviation). IPW = inverse probability of weighting; ASD = absolute standardized mean difference; FAR = fibrinogen/albumin ratio; ICU = intensive care unit.

**Table 3 jcm-12-01407-t003:** Clinical outcome according to the estimated threshold of fibrinogen/albumin ratio of 10.79.

	Low Group	High Group	Unadjusted HR (95% CI)	*p* Value	IPW Adjusted HR (95% CI)	*p* Value
	(*n* = 12,533)	(*n* = 6029)				
One-year mortality	1551 (12.4)	2195 (36.4)	3.47 (3.25–3.70)	<0.001	1.72 (1.59–1.86)	<0.001
30-day mortality	955 (7.6)	1152 (19.1)	2.68 (2.46–2.92)	<0.001	1.32 (1.19–1.46)	<0.001
In-hospital mortality	1098 (8.8)	1313 (21.8)	2.74 (2.52–2.96)	<0.001	1.33 (1.21–1.47)	<0.001
ICU mortality	614 (4.9)	677 (11.2)	2.42 (2.17–2.70)	<0.001	1.10 (0.97–1.26)	0.140

IPW, inverse probability of weighting; HR, hazard ratio; CI, confidence interval, ICU, intensive care unit.

## Data Availability

Regarding data availability, our data are available on the Harvard Dataverse Network (https://doi.org/10.7910/DVN/NAW23F (accessed on 9 January 2023)) as recommended repositories.

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
