# Peer review of "Association between Fibrinogen-to-Albumin Ratio and Prognosis in Patients Admitted to an Intensive Care Unit"

_jcm, 2023, doi:10.3390/jcm12041407_

Round 1

Reviewer 1 Report

The authors performed a retrospective study investigating associations between fibrinogen to albumin-ratio (FAR) and 1-year mortality in intensive care unit (ICU) patients. A large number of patients (n= 18,562) with different underlying conditions (medical, surgical, oncological) were included. The authors calculated area under ROC curves and adjusted HR for 1-year mortality for both FAR and SOFA score. They found moderate ability of FAR to predict 1-year mortality (area under ROC curve 0.68), adjusted HR 1,74 (95%CI 1.59–1.86)). FAR was not superior to SOFA score overall, but when patients were grouped according to underlying conditions, there were some differences.

Investigating new biomarkers for clinical outcomes are of interest. Strengths of the study is the large number of patients and availability of clinical data which allows for adjustment. Limitations is the retrospective nature and especially the fact that fibrinogen measurements were available for only 30% of the ICU patients in the database (see my comment below).

The rationale behind choosing FAR as a biomarker which is explained in the introduction and discussion is a little vague.

I have the following specific comments:

Methods section:

1) Have you measured FAR in a group of healthy individuals of same demographics as your patients?

2) Regarding SOFA score, were all relevant variables available for all patients in the medical record, including FiO2, vasopressor, and GCS?

3) Why did you choose 1-year mortality and not a shorter interval, e.g. 30-day mortality, ICU mortality or in-hospital mortality? The latter may be even more important for clinical decision making. If you have the data for these outcomes, I suggest that they are added to the manuscript.

Results section:

4) Less than 30% of the patient in the database had fibrinogen measured. Therefore, there is obviously some doubt of the external validity of the results, as the patients who had a fibrinogen measurement at admission may differ from those who hadn’t – more severe illness/bleeding/suspicion of DIC etc.

5) Demographic and clinical characteristics displayed in Table 1: The tertile groups appear different regarding several clinical conditions, e.g. malignancy, chronic kidney and liver disease, recent surgery, respiratory distress and cardiovascular disorder, all of which could influence the results. This should be commented on.

Conclusion:

6) You state that #Therefore, FAR is feasible and reliable in predicting long-term mortality in these patients.” I don’t think that an area under ROC curve of 0.68 and an adjusted HR of 1,74 justifies this conclusion.”

Minor: Table 1, mechanical ventilation, 1st tertile: a parenthesis is missing?

Author Response

Response to Reviewers

Reviewer #1:

The authors performed a retrospective study investigating associations between fibrinogen to albumin-ratio (FAR) and 1-year mortality in intensive care unit (ICU) patients. A large number of patients (n= 18,562) with different underlying conditions (medical, surgical, oncological) were included. The authors calculated area under ROC curves and adjusted HR for 1-year mortality for both FAR and SOFA score. They found moderate ability of FAR to predict 1-year mortality (area under ROC curve 0.68), adjusted HR 1,74 (95%CI 1.59–1.86)). FAR was not superior to SOFA score overall, but when patients were grouped according to underlying conditions, there were some differences.

Investigating new biomarkers for clinical outcomes are of interest. Strengths of the study is the large number of patients and availability of clinical data which allows for adjustment. Limitations is the retrospective nature and especially the fact that fibrinogen measurements were available for only 30% of the ICU patients in the database (see my comment below).

The rationale behind choosing FAR as a biomarker which is explained in the introduction and discussion is a little vague.

  1. We appreciate forethoughtful comments for our study. Although we have suggested that FAR may be useful for ICU patients, indications for its use have not yet been established. It may not be necessary to measure fibrinogen and albumin in all patients upon ICU admission, as this would likely be resource-intensive and may not provide significant additional prognostic information for all patients. Instead, it may be more appropriate to use a risk stratification approach to identify patients at high risk of poor outcomes who would most benefit from measuring these biomarkers. In this study, fibrinogen was routinely checked on ICU admission in patients with sepsis, suspicious disseminated intravascular coagulation, solid and hematologic malignancies, post-operative care, suspicious bleeding tendency, etc. Additionally, measuring fibrinogen and albumin in patients who have already been identified as being at high risk of poor outcomes based on other clinical or laboratory parameters may also be useful in predicting prognosis.

We added the above sentences in the Discussion section of revised manuscript (line 213-221, page 15-16).

I have the following specific comments:

Methods section:

1) Have you measured FAR in a group of healthy individuals of same demographics as your patients?

  1. In this study, we included only the patients treated at ICU and with available SOFA score, fibrinogen, and albumin levels at ICU admission. Also, this was a retrospective study, so we have not measured FAR in healthy people. 

2) Regarding SOFA score, were all relevant variables available for all patients in the medical record, including FiO2, vasopressor, and GCS?

  1. In our practice, the SOFA score has been measured at the time of ICU admission in critically ill patients and we included only the patients with SOFA score for this study. The SOFA score was not calculated separately for this study, and we just used the score recorded at the time of ICU admission. Therefore, there were no missing data for factors to measure SOFA score such as FiO2, use of vasoactive agents and GCS.

3) Why did you choose 1-year mortality and not a shorter interval, e.g. 30-day mortality, ICU mortality or in-hospital mortality? The latter may be even more important for clinical decision making. If you have the data for these outcomes, I suggest that they are added to the manuscript.

  1. We agree reviewer’s comment. In this study, we tried to investigate long-term mortality. The objective of this study was to investigate the usefulness of FAR as a long-term prognostic marker in patients admitted to an ICU for various causes compared with SOFA score, a widely used prognostic scoring system. The SOFA score, which directly reflects organ failure, was much better than FAR in predicting 30-day mortality and ICU mortality. We added 30-day mortality, ICU mortality, and hospital mortality according to FAR level in Table 3 and revised manuscript as follows.

The primary endpoint was mortality during 1-year follow-up. The secondary endpoints were 30-day mortality, in-hospital mortality, and ICU mortality. (line 64-65, page 6)

We added following the results in the revised manuscript (line 115-124, page 10).

The risk of one-year mortality was increased for the high FAR group compared with the low FAR group (36.4% vs. 12.4%; HR: 3.47; 95% CI: 3.25–3.70; p < 0.001 for 1-year mortality). 30-day mortality, in-hospital mortality, and ICU mortality were also higher in the high FAR group (19.1% vs. 7.6%; HR, 2.68; 95% CI, 2.46–2.92; p < 0.001 for 30-day mortality, 21.8% vs. 8.8%; HR, 2.74; 95% CI, 2.52–2.96; p < 0.001 for in-hospital mortality, and 11.2% vs. 4.9%; HR, 2.42; 95% CI, 2.17–2.70; p < 0.001 for ICU mortality). After adjustment with IPW technique, the association between increased risk of mortality and high FAR remained significant except for ICU mortality (HR, 1.72; 95% CI, 1.59–1.86; p < 0.001 for 1-year mortality, HR, 1.32; 95% CI, 1.19–1.46; p < 0.001 for 30-day mortality, HR, 1.33; 95% CI, 1.21–1.47; p < 0.001 for in-hospital mortality) (Table 3).

Table 3. Mortality according to the estimated threshold of fibrinogen/albumin ratio of 10.79

Low group

High group

Unadjusted HR (95% CI)

P value

IPW adjusted HR (95% CI)

P value

(N=12,533)

(N=6,029)

One-Year follow-up

1551 (12.4)

2195 (36.4)

3.47 (3.25-3.70)

<0.001

1.72 (1.59-1.86)

<0.001

30-day follow-up

955 (7.6)

1152 (19.1)

2.68 (2.46-2.92)

<0.001

1.32 (1.19-1.46)

<0.001

In-hospital mortality

1098 (8.8)

1313 (21.8)

2.74 (2.52-2.96)

<0.001

1.33 (1.21-1.47)

<0.001

ICU mortality

614 (4.9)

677 (11.2)

2.42 (2.17-2.70)

<0.001

1.10 (0.97-1.26)

0.140

IPW, inverse probability of weighting; HR, hazard ratio; CI, confidence interval

Results section:

4) Less than 30% of the patient in the database had fibrinogen measured. Therefore, there is obviously some doubt of the external validity of the results, as the patients who had a fibrinogen measurement at admission may differ from those who hadn’t – more severe illness/bleeding/suspicion of DIC etc.

  1. We fully agree with your opinion that our results may not be generalized to other populations or settings. Specifically, the fact that less than 30% of patients in the database had fibrinogen measured raises concerns about whether the sample is representative of the larger population of patients with similar conditions. Following your comments, we mentioned it as a limitation of this study and suggested the need for further studies in the Discussion section as follows.

“In this study, only patients with fibrinogen available at ICU admission were included. This could raise concerns about selection bias and external validity of our findings. Additionally, they may have had more severe and complex disease, potentially leading to an overestimation of mortality. Nevertheless, our results showed consistent trend with previous studies, which demonstrated an association between high FAR and mortality risk in ICU patients (Ref 33, 34). Future studies with larger sample size, and with measurements of fibrinogen in all patients could provide further support for our findings.” (line 206-212, page 15)

Ref 33. AfÅŸin, A.; Tibilli, H.; HoÅŸoÄŸlu, Y.; AsoÄŸlu, R.; Süsenbük, A.; Markirt, S.; Tuna, V.D. Fibrinogen-to-albumin ratio predicts mortality in COVID-19 patients admitted to the intensive care unit. Adv Respir Med 2021. https://doi.org/10.5603/ARM.a2021.0098.

Ref 34. Bender, M.; Haferkorn, K.; Tajmiri-Gondai, S.; Uhl, E.; Stein, M. Fibrinogen to Albumin Ratio as Early Serum Biomarker for Prediction of Intra-Hospital Mortality in Neurosurgical Intensive Care Unit Patients with Spontaneous Intracerebral Hemorrhage. J Clin Med 2022, 11. https://doi.org/10.3390/jcm11144214.

5) Demographic and clinical characteristics displayed in Table 1: The tertile groups appear different regarding several clinical conditions, e.g. malignancy, chronic kidney and liver disease, recent surgery, respiratory distress and cardiovascular disorder, all of which could influence the results. This should be commented on.

  1. We agree reviewer’s comment. As your recommendation, we have revised Table 1 with p-values to specify that the results differed in comorbidity. We add the following sentence in Results section of revised manuscript (line 97-98, page 7).

There were statistical differences in most variables between 3 groups, except severe trauma.

Conclusion:

6) You state that #Therefore, FAR is feasible and reliable in predicting long-term mortality in these patients.” I don’t think that an area under ROC curve of 0.68 and an adjusted HR of 1,74 justifies this conclusion.”

  1. We agree with reviewer’s comment that we may have over-interpreted our results. We revised conclusions as the below sentences (line 239-242, page 16-17).

In this study, FAR and SOFA score at ICU admission were associated with 1-year mortality in patients admitted ICU. However, obtaining FAR is easier than obtaining SOFA score in critically ill patients. Therefore, FAR is feasible and might help predict long-term mortality in these patients.

Minor: Table 1, mechanical ventilation, 1st tertile: a parenthesis is missing?

  1. We apologize for this error. We revised table 1.

We appreciate forethoughtful comments for our study. Addressing them fully has significantly strengthened the manuscript.

Reviewer 2 Report

This is a well writtten paper.

Please let me make sure about the following points.

What is new in this research? Please emphasize the novely of this research.

How did you deal with patients who had transfusion or albumin infusion? Did you exclude all?

Elderly patients generally have lower albumin level. Do you think the age affects in your research?

You need to do subanalysis in patients who have hemmorrhage because it severely affects the fibrinogen level.

How do you think this research contributes to patients?

What is the relationship with physical function with Fibrinogen-to-Albumin Ratio ?

Do you think we need to measure fibrinogen and albmin in all patients. If no, you need to explain what case we need to measure these biomarkers to predit prognosis.

In this database, what is the standard to measure fibrinogen? Because it is not routine to measure coagulation, you need to show the clinical practice when these biomarkers were meausred.

Author Response

Response to Reviewers

Reviewer #2:

This is a well written paper.

Please let me make sure about the following points.

What is new in this research? Please emphasize the novelty of this research.

  1. First, we thank the reviewer for valuable comments. Recently, some studies have shown its usefulness and significance as a prognostic factor in several diseases including cancer, coronary artery disease, stroke-associated pneumonia, and sepsis. However, there was no study for usefulness of FAR in the entire ICU. Therefore, our study seems to be an advantage in generalizability. In addition, it differed from other studies in that FAR was analyzed relative to SOFA scores and analyzed separately for each ICU in this study. As you recommended, we mentioned the novelty of our study in the Discussion section as follows.

Despite these limitations, this is the first study to evaluate the usefulness of FAR in the entire ICU. Additionally, our study differed from other studies in that FAR was analyzed relative to SOFA scores and analyzed separately for each ICU. In this context, the present study provides valuable insights, but large-scale prospective studies are needed to further confirm the usefulness of FAR in predicting clinical outcomes of critically ill patients with evidence-based conclusions. (line 231-236, page 16)

How did you deal with patients who had transfusion or albumin infusion? Did you exclude all?

  1. In this study, albumin level was based on the initially checked albumin value after ICU admission. In general, albumin infusion is considered based on the initial albumin value, therefore, it is assumed that the value before albumin infusion was used in the study. Transfusion was not fully investigated, but the first checked value of the fibrinogen was also used. There was a possibility of fibrinogen change by transfusion, and this was additionally described in the limitations of the revised manuscript (line 226-228, page 16).

Although fibrinogen and albumin levels were based on initially identified values to reduce the changes due to blood transfusions and albumin infusions after ICU admission, the values might have been affected in some patients.

Elderly patients generally have lower albumin level. Do you think the age affects in your research?

  1. We agree with reviewer’s comment. In statistical method of this study, we chose an inverse probability weighting using propensity score and adjusted for all relevant variables in this study. Logistic regression was used in the process of setting the propensity score. Therefore, it could be seen that the effect of age was statistically reflected to some extent.

You need to do subanalysis in patients who have hemorrhage because it severely affects the fibrinogen level.

  1. We agree with reviewer’s comment. We could not find the patients with hemorrhage because of retrospective nature of this study. Therefore, we investigated the patients with transfusion of more than 3 packs of RBC. In sub-analysis according to transfusion, FAR may be a better predictor of mortality in RBC transfused patients with less than 3 packs compared with those with 3 packs or more. However, the tendency for patients with higher FAR values to have a worse prognosis was similar in both transfused populations.

Transfusion less than 3 packs of RBC

Low group

High group

Unadjusted HR (95% CI)

P value

(N=6548)

(N=2550)

One-Year follow-up

408 (6.2)

640 (25.1)

4.72 (4.17-5.35)

<0.001

30-day follow-up

282 (4.3)

403 (15.8)

3.99 (3.42-4.64)

<0.001

In-hospital mortality

286 (4.4)

378 (14.8)

3.70 (3.18-4.32)

<0.001

ICU mortality

181 (2.8)

244 (9.6)

3.67 (3.03-4.45)

<0.001

Transfusion of more than 3 packs of RBC

Low group

High group

Unadjusted HR (95% CI)

P value

(N=5985)

(N=3479)

One-Year follow-up

1143 (19.1)

1555 (44.7)

2.71 (2.51-2.93)

<0.001

30-day follow-up

673 (11.2)

749 (21.5)

2.01 (1.81-2.23)

<0.001

In-hospital mortality

812 (13.6)

935 (26.9)

2.15 (1.96-2.36)

<0.001

ICU mortality

433 (7.2)

433 (12.4)

1.79 (1.57-2.05)

<0.001

How do you think this research contributes to patients?

  1. Since the FAR is a relatively simple test that can be performed quickly and at low cost, it may be more practical to use in the early stages of ICU admission. Additionally, it could be useful in identifying patients who are at high risk of death and in need of more aggressive treatment. However, the SOFA score is a more comprehensive tool that takes into account multiple organ systems. Therefore, it would be important to use both scores in conjunction with each other, as well as other clinical data, in order to make the best decisions for patient care.

What is the relationship with physical function with Fibrinogen-to-Albumin Ratio ?

  1. We investigated functional capacity as physical function in study population. We defined as functional limitation as Functional capacity ≤ 4METs (Metabolic equivalents (METS) in exercise testing, exercise prescription, and evaluation of functional capacity).

FAR was higher in patients with functional limitation than those without functional limitation (p < 0.001). In addition, functional limitation more common in high FAR group than in low FAR group.

Total

No functional limitation (N =14923)

Functional limitation (N =4018)

P

Fibrinogen

289.4 (163.4)

372.8 (195.9)

<0.001

Albumin

3.18 (0.66)

2.93 (0.62)

<0.001

FAR

9.47 (6.06)

13.19 (7.50)

<0.001

Total

Low group (N =12533)

High group (N =6029)

P

Functional limitation

1648 (13.1)

1991 (33.0)

<0.001

Do you think we need to measure fibrinogen and albumin in all patients. If no, you need to explain what case we need to measure these biomarkers to predict prognosis.

  1. We agree with your opinion that we need to clarify the indication of measurement of fibrinogen and albumin in ICU patients. Following your recommendation, we described which patients could benefit from measuring FAR in the Discussion section as follows.

Although we have suggested that FAR might be useful for ICU patients, indications for its use have not yet been established. It may not be necessary to measure fibrinogen and albumin in all patients upon ICU admission, as this would likely be resource-intensive and may not provide significant additional prognostic information for all patients. Instead, it may be more appropriate to use a risk stratification approach to identify patients at high risk of poor outcomes who would most benefit from measuring these biomarkers. Additionally, measuring fibrinogen and albumin in patients who have already been identified as being at high risk of poor outcomes based on other clinical or laboratory parameters may also be useful in predicting prognosis. (line 213-221, page 15-16)

In this database, what is the standard to measure fibrinogen? Because it is not routine to measure coagulation, you need to show the clinical practice when these biomarkers were measured.

  1. In our practice, fibrinogen was routinely checked on ICU admission in patients with sepsis, suspicious disseminated intravascular coagulation, solid and hematologic malignancies, post-operative care, suspicious bleeding tendency, etc. In this study, only patients with fibrinogen available at ICU admission were included. This could raise concerns about selection bias and external validity of our findings. Additionally, they may have had more severe and complex disease, potentially leading to an overestimation of mortality. Nevertheless, our results showed consistent trend with previous studies, which demonstrated an association between high FAR and mortality risk in ICU patients. Future studies with larger sample size, and with measurements of fibrinogen in all patients could provide further support for our findings.

As your recommendation, we added the following sentences in the Methods section of the revised manuscript (line 50-53, page 5-6)

The decision to measure fibrinogen levels was based on the individual patient's clinical situation. Fibrinogen was routinely checked on ICU admission in patients with sepsis, suspicious disseminated intravascular coagulation, solid and hematologic malignancies, post-operative care, suspicious bleeding tendency, etc.

We thank the reviewer for valuable recommendation. Addressing them fully has significantly strengthened the manuscript.

Round 2

Reviewer 1 Report

The authors have responded satisfactorily. I have one remaining comment:

In your reply regarding my question on SOFA score, you state that you included only patients with SOFA score available. How many patients did not have available SOFA score? This percentage should be stated in the methods section. If all patients had available SOFA score, then it could be mentioned in the methods section, e.g. "SOFA score was reported in the medical record for all included patients" or something to this effect.

Author Response

Response to Reviewers

Reviewer #1:

In your reply regarding my question on SOFA score, you state that you included only patients with SOFA score available. How many patients did not have available SOFA score? This percentage should be stated in the methods section. If all patients had available SOFA score, then it could be mentioned in the methods section, e.g. "SOFA score was reported in the medical record for all included patients" or something to this effect.

  1. Thanks for your valuable comments. As mentioned in previous rebuttal letter, the SOFA score has been measured at the time of ICU admission in critically ill patients and we included only the patients with SOFA score for this study. In our hospital, a SOFA score has been measured after ICU admission in all ICU patients. In addition, the SOFA score was not calculated separately for this study, and we just used the score recorded at the time of ICU admission. Finally, all patients had available SOFA score in this study.

As your recommendation, we added the following sentence in the Methods section of the revised manuscript (line 39-40, page 5)

In this study, the SOFA score was reported in the medical record for all included patients.

We appreciate forethoughtful comments for our study. Addressing them fully has significantly strengthened the manuscript.

Reviewer 2 Report

Authors responded sufficiently and very politely to my review.

Author Response

Sincere thanks for your comment. We appreciate forethoughtful comments for our study. Addressing them fully has significantly strengthened the manuscript.